# Autism Spectrum Disorder in the Dominican Republic: A Mini Review of the Current Situation

**DOI:** 10.3390/children10010121

**Published:** 2023-01-06

**Authors:** Yosauri Fernandez Figuereo, Jack Lewis, Peyton Lee, Stephen J. Walker

**Affiliations:** 1Department of Molecular Genetics and Genomics, Wake Forest University Graduate School of Arts and Sciences, Winston Salem, NC 27101, USA; 2Reynolda Campus, Wake Forest University, Winston Salem, NC 27109, USA; 3Wake Forest Institute for Regenerative Medicine, Winston Salem, NC 27101, USA

**Keywords:** autism in the DR, autism, autism spectrum disorder, Dominican Republic, low-income country, Latin America

## Abstract

As the recognition of autism spectrum disorder (ASD) increases, and the prevalence estimates of ASD continue to rise throughout the world, it has become apparent that access to diagnostic and treatment services is highly dependent on geography. Even within countries such as the United States, which has received significant interest and investment in understanding, diagnosing, treating, and providing programs for those with ASD over the last 20+ years, access to information and services is uneven. In poorer countries such as the Dominican Republic (DR), where >40% of citizens live below the poverty level and access to quality healthcare overall continues to be a challenge, issues associated with ASD are not yet being adequately addressed. The objective of this review is to provide a realistic synopsis of the resources currently available to Dominicans who have a family member or loved one with ASD. We examine the challenges these families face in finding care, the stigma associated with ASD, and programs available for people with ASD. We conclude that while the DR is making progress in its efforts to address ASD, there is still much work to be done.

## 1. Introduction

In 2013, the American Psychiatric Association (APA) integrated autistic disorder, childhood disintegrative disorder, pervasive developmental disorder-not-otherwise specific (PDD-NOS), and Asperger’s syndrome into one concept known today as autism spectrum disorder (ASD). ASD is a multifactorial syndrome that impacts the affected child’s neurodevelopmental and behavioral progress. Clinically, the APA defines ASD as a “complex developmental condition involving persistent challenges with social communication, restricted interests, and repetitive behavior” [1]. The degree of behavioral and cognitive impact in persons with ASD varies; affected individuals display a variety of manifestations depending on their unique environmental and genetic factors. 

The Diagnostic and Statistical Manual of Mental Disorders (DSM-5) describes the cardinal signs and symptoms of ASD as: difficulty interacting and communicating, lack of interest, repetitive behaviors, and difficulty learning and functioning in school or other areas that request focus [2]. An individual can be diagnosed with ASD at any age; however, the signs of ASD usually appear during the early stages of childhood. Studies have shown that early diagnosis and intervention provides children with ASD a higher probability of developing to their full potential [3]. Therefore, broad access to both basic and specialized treatment and services is crucial to ensure the best outcomes for children with ASD.

In countries such as the United States, where a relatively advanced knowledge of the topic has been achieved, access to data regarding ASD is readily available. However, in countries with socioeconomic challenges, such as the Dominican Republic (DR), access to information and critical data is more difficult to attain, or even non-existent. For example, there is no accurate record of the number of children with ASD and access to services is limited, especially for low-income families. It is likely that there are more children with ASD than has actually been reported in the DR. Therefore, the aim of this narrative review is to provide an overview of the current situation in the DR for individuals with ASD, and to discuss the emergent need for resources that could aid in early diagnoses, treatment, educational opportunities, and access to long-term care.

## 2. Materials and Methods

An electronic database search was conducted in the National Library of Medicine, the National Center for Biotechnology Information website (PubMed.gov). In these searches, the following keywords and phrases were used: autism in the DR, services, autism, autism spectrum disorder, Latin America, Dominican Republic, and low-income families. Retrieved documents were analyzed and organized based on the following inclusion criteria: the articles, in English, were published in a peer-reviewed journal and contained relevant information about group statistics, available services, and the financial situations of families with ASD in the DR. An electronic database search was also conducted from governmental and non-governmental institutional websites in the DR which are presented exclusively in Spanish. Spanish-language transcriptions were read, reviewed, and translated by the lead author who is bilingual, bicultural, and originally from the DR. These translations were then reviewed by the coauthors. This information was used to populate data presented in results.

## 3. Results

### 3.1. Current Situation

#### 3.1.1. Prevalence of ASD in the DR

A recent survey by the Latin American Autism Spectrum Network (Red Espectro Autista Latinoamerica network, a coalition of researchers/clinicians from six Latin American countries) reported on the needs and challenges faced by individuals with autism in Latin America, including the DR. According to the document, there are approximately six million individuals living with ASD in Latin America. The results showed that long wait times, cost of treatment, and lack of specialized services were the main challenges, followed by stigma, financial problems, and time constraints. According to the survey, individuals with ASD in the DR were less verbal and used private services exclusively, compared to the other countries (Argentina, Brazil, Chile, Uruguay, and Venezuela). Furthermore, Dominican families experienced significantly more financial issues resulting in more complications in paying for specialized treatment [4]. This document supports the hypothesis that there are many more individuals with ASD in the DR than have been previously reported—likely due, at least in part, to the fact that most patients come from families that cannot pay for private services, and therefore cannot be registered.

There are no official records regarding the number of individuals living with ASD in the DR. The Fundación Dominicana del Autismo (FDA) estimates, based on the 2010 National Census, that 12.3% of the population has some form of disability [5]. Using the data reported in 2018, the Center for Disease Control (CDC) estimates that 1 out 88 children (1.14%) in the US are diagnosed with ASD [6]. In the DR, this would translate to approximately 107,676 individuals, representing 9.28% of people with disability. Multiple studies have found that ASD is more common in boys than girls; the DR follows the same pattern, where for every nine children with ASD, one is female. Furthermore, the CDC highlights that most children are diagnosed with ASD after six years of age, which limits their potential for typical development [5]. The American Academy of Pediatrics recommends that children be screened for autism at 18 months of age to enable early intervention [2]. As of now, there are presumed to be thousands of undiagnosed cases in the DR [5].

#### 3.1.2. Access to Services

The availability of scientific data describing the accessibility and quality of special services required for individuals with ASD in Latin America is still limited [7]. Many services do not meet the lifespan needs of these individuals as they receive very few hours of treatment, if any at all, after transitioning from childhood to adulthood [8]. Multiple factors impact the accessibility of specialized services for ASD individuals in Latin America including, but not limited to, the need for political reform to address the limitation of available mental health services [9]. In addition, studies have shown that stigmatization could play an important role in how caretakers approach services in Latin countries where caregivers of adult individuals with ASD and female children are stigmatized to a greater degree than other groups among the ASD population [10].

Overall, the healthcare system in the DR is insufficient to address the needs of the population. Moreover, people with autism have an even higher probability of not receiving adequate treatment from the current healthcare system. In 2008, a report on the assessment of mental health systems in the DR revealed that less than 1% (0.38%) of the national budget goes toward mental health services [11]. From the three main sectors in the DR health system: public, private, and social insurance [12], private health care is often what covers care for ASD individuals [4]. Most of these private institutions are nonprofit organizations founded by parents of children with ASD (Table 1). Social insurance does not cover most of the treatment for ASD individuals, representing a significant challenge for low-income families.

#### 3.1.3. Legal Framework

In the DR, the government has a contractual obligation to protect the rights of individuals with special needs. The DR ratified the United Nations Convention on the Rights of Persons with Disabilities (CRPD) in 2009 [20]. The purpose of the CPRD is “to promote, protect, and ensure the full and equal enjoyment of all human rights and fundamental freedoms by all persons with disabilities, and to promote respect for their inherent dignity [21]”. Subsequently, the Law on Equal Rights of Persons with Disabilities No. 5–13 was approved in 2015, published in 2016, and is designed to protect the rights of people with disabilities and promote their effective inclusion into society. Most of the institutions that work for the benefit of ASD individuals use this law as a reference for their operations [22].

#### 3.1.4. Public Awareness

In rural areas within developing nations such as the DR, the population often has a very limited knowledge of mental health disorders and neurodevelopmental conditions. The lack of awareness regarding autism among the Dominican population is one of many obstacles that can hinder progress in receiving adequate diagnosis and treatment. Moreover, this lack of understanding about ASD can exacerbate the problem due to misconceptions (e.g., “nothing can be done to help people with ASD”) and stigmatization [23]. Other countries, such as India, have identified similar problems where often these conditions are difficult to accept for families due to misconceptions [24]. In low- and middle-income countries, most parents do not have the resources to identify early signs of ASD which can cause a delay in seeking a formal diagnosis and potential treatment for their children [25]. In recent years, Dominican nonprofit organizations have been actively advocating in favor of ASD awareness on radio and TV, and supporting the approval of a specific legal framework that protects the right of individuals with ASD.

#### 3.1.5. Educational Services

Educational opportunities for children with ASD are severely lacking in the Dominican Republic. There are no records of public educational institutions for children with ASD and other neurodevelopmental conditions in the country. This means low- and middle-income families have limited access to schooling for their ASD children. This situation is even more difficult for children with severe ASD due to the need for schools with a specially equipped infrastructure and trained staff. Studies also suggest that, among individuals with developmental disorders, people with ASD are more likely to be discriminated against and excluded [26]. In turn, this leads to increase stigmatization and segregation.

## 4. Recommendations

The efforts of the government and relevant healthcare authorities in the DR should be focused on building a solid platform throughout the country to help families at all socioeconomic levels receive appropriate services to address the needs of ASD individuals. This review emphasizes that early diagnosis, healthcare and specialty care centers, public awareness, and an integral education should be the key elements for a primary course of action.

### 4.1. Early Diagnosis

According to the Early Start Denver Model (ESDM), “a comprehensive developmental behavioral intervention for improving outcomes of toddlers diagnosed with autism spectrum disorder [27]”, early detection and intervention is one essential element. Using this framework, toddlers with ASD between 18 and 30 months of age who received ESDM treatment showed a significant improvement in IQ, adaptative behavior, and autism diagnosis compared to children who received community-based intervention [28]. To be effective, an early diagnosis of ASD must be accompanied by proper treatment. There is evidence to suggest that in addition to early diagnosis, children with optimal outcomes also received intensive behavioral intervention and less pharmacologic treatment [29].

In low-income countries such as the DR, individuals with ASD may receive a more delayed diagnosis compared to individuals in more affluent countries such as the United States. Clinical characteristics and socioeconomic factors are the leading influences on receiving a later diagnosis [30]. Parents are usually the ones closest to ASD children, so they must be instructed to recognize the early signs and look for appropriate help. The government should create educational strategies that involve parents, especially for families in non-urban areas who do not have a clear concept of ASD. Information and support should be provided to them in convenient locations such as local hospitals and public schools. Parents and caregivers should be provided with detailed information about ASD and where they should go for treatment and educational services [31]. Moreover, parents that receive proper training can serve as educators to counterbalance the non-professional approach most commonly seen in developing countries [32]. This could help to not only improve the outlook for children with ASD, but also to eliminate some of the stigma attached to autism-related behaviors.

In America and other countries that have devoted significant resources to the understanding and diagnosis of ASD, validated questionnaires have been developed such as the Autism Diagnostic Observation Schedule (ADOS) and the Autism Diagnostic Interview-Revised (ADI-R). These are both multi-step and multi-question exams used by more affluent countries to diagnose ASD and are considered “…the gold standards to facilitate diagnosis of ASD in High-Income Countries (HICs)”. However, these tests are expensive, time-consuming, and require certifications to administer and interpret [33].

Jamaica serves as a model for a lower-income country utilizing an alternative to the ADOS and ADI-R questionnaires. Jamaica has begun to utilize a test called the Childhood Autism Rating Scale (CARS). The CARS evaluation takes only 5–10 min to administer and score while the ADOS and ADI-R questionnaires can take up to an hour [33]. This scale has seen so much success that some HICs such as the United States are also using it as an additional tool to assess ASD. The CARS evaluation is also significantly more cost effective than the ADOS and ADI-R; it costs ~350 USD to obtain test booklets for 100 children while the ADOS can cost >2000 USD to obtain test booklets for the same number of children. Furthermore, in a study conducted in Jamaica, children aged 2–8 years (*n* = 149) were initially diagnosed with ASD using CARS. When these same children were re-evaluated with ADOS and ADI-R tests, the diagnostic agreement between CARS and ADOS was 100%, while agreement between CARS and ADI-R was 94.6% [33]. The effectiveness of CARS as a cheaper and quicker alternative diagnostic tool bodes well for the implementation in the DR. The CARS test is a great way for parents with concerns about their children to get a quick and affordable diagnostic that can help determine if their child has ASD.

A study from Brazil presents evidence that more conventional approaches to diagnose ASD, such as the Diagnostic and Statistical Manual of the American Psychiatric Association (DSM) and the CARS, represent a stress factor for families of ASD individuals due to diagnostic postponement. Most families experience difficulties in the identification of early signs of ASD, and with limited access to healthcare, the postponement of a diagnosis compounds the stress of the individual and the family. To improve the care of ASD individuals, in 2013, the Brazilian Ministry of Health published Guidelines for the Care and Rehabilitation of Individuals with ASD, to provide caregivers and family a tool for early identification of ASD. Despite these efforts, the study concluded that these guidelines had been poorly implemented and proved that more specialists needed to be involved to ensure its effectiveness [34]. In s study from Tunisia, in which a semi-structured interview was performed with eight mothers of preschool children with ASD demonstrated that all the children were diagnosed by experienced psychiatrists following DSM guidance. This study further provides evidence that after children were diagnosed with ASD, parents experienced intense distress that goes on to be a detriment to their emotional and physical health. This situation tends to improve as parents accept their children’s condition over time [35]. In China, a survey of parents of ASD children revealed that low levels of education, often associated with low-income communities, correlated with a late diagnosis of ASD [36]. The implementation of effective and affordable testing for early detection of ASD in low-income countries could not only improve the quality of life of ASD individuals, but the overall well-being of their families.

### 4.2. Healthcare and Specialty Care Centers

Ensuring that individuals with ASD have access to public institutions with effective treatment and personnel is critical for successful implementation. Once parents can detect early signs of autism, they should be able to access a free screening for ASD to determine if their child needs immediate attention. The DR does not have enough public institutions for children with disabilities, which makes it especially difficult for families in non-urban areas to access specialty services. Most of the urban and non-urban regions of the DR have a local hospital that provides primary healthcare to adults and infants. Each of these hospitals should collaborate with specialists for screening, diagnosis, and treatment of ASD, and other children with disabilities. 

In addition, ASD individuals should have healthcare coverage throughout their whole life. The few public institutions that treat ASD children in the DR are designed to provide services to children only from 1 to 6 years old, leaving adolescents and adult individuals with limited treatment options. A recent study showed that in several countries from Latin America (including the DR), approximately 86% to 95% of adults obtain zero hours of health care services weekly [9]. The low quality of services and lack of time families are allotted for treatment compounds the problem even further. Effective community-delivery programs, such as the World Health Organization (WHO)-led Caregiver Skill Training (CST) and parent-mediated intervention for autism spectrum disorders in South Asia (PASS) Plus caregiver training program have been established in lower resource countries and have amplified access to healthcare services for individuals with ASD [37,38,39]. These programs can be adapted and modified for effective application in the DR.

### 4.3. National Awareness

Awareness and proper capacitation for the detection of early signs of ASD are crucial. The government should redirect efforts to educate the population regarding ASD, especially in rural areas of the country. This is a difficult issue as there is often a lack of education, confusion, and correlation of autism with other mental illnesses. Studies suggest that a focused educational campaign could facilitate the implementation of an efficient system to enhance awareness among the population [40,41]. More studies should be conducted throughout the country to determine the level of knowledge of the population regarding ASD. This will help to determine what areas are in greatest need for proper diagnosis and treatment. This campaign should be directed toward all levels: the public, caregivers, parents, and physicians, since studies suggest that the level of misconception and obliviousness regarding ASD are not restricted to any single demographic [42,43].

### 4.4. Integral Education

Along with a proper awareness campaign, an integral education system should be instituted around the country to provide all families with the opportunity to integrate their ASD children into an inclusive educational environment. For children with severe ASD, special institutions should be established around the country with the capacity to cover the needs of this at-risk population. An inclusive classroom environment is needed to ensure effective academic and social integration of children with ASD [44,45], and appropriately trained teachers are essential in promoting that inclusivity within education [46]. Studies have also emphasized that communication and collaboration between parents and school staff is a pivotal element to ensure academic and social inclusivity for ASD children [44]. A recent study in South Africa, where interviews were made with multi-sectoral ASD service providers in Western Cape Province, found that educator training, parent-mediated early intervention, and intersectoral and inter-professional collaborations were the key factors in delivering effective educational services to children with ASD [25]. In addition, the Autism Society promotes the application of an Individual Education Plan (IEP), an educational program that adjusts to each child’s specific abilities and needs [47].

## 5. Conclusions

Access to affordable medical, educational, and support services, anchored by a solid legal framework, is necessary to address the current and future needs of individuals with ASD in the DR. The effectiveness of any public policy structure directed towards addressing these needs is dependent on the appropriate implementation (to provide public institutions the tools they need to provide the necessary resources), and follow-up (to ensure compliance). In the DR, priority should be given to increasing awareness and understanding of ASD, providing easier access to diagnostic, treatment, and education opportunities, regardless of socioeconomic status, and to providing support that extends beyond childhood for persons with ASD.

## Figures and Tables

**Table 1 children-10-00121-t001:** Dominican ASD organizations that were found with an electronic search and have an official website for public access [5,13,14,15,16,17,18,19]. The information on the websites is in Spanish. Spanish-language transcriptions were read and reviewed by the lead author who is bilingual, bicultural, and originally from the DR.

Name	Year	City	Initiative	Category	Status
Fundación Dominicana de Autismo-FDA (Dominican Autism Foundation)	1995	Santo Domingo, D. N.	Parents	Nonprofit Organization	Active
Centro de Terapia Integral para Niños Autistas—TINA	1967	Santiago	^1^ INA	Nonprofit Organization	Active
Fundación Manos Unidas por Autismo-FMUA (Manos Unidas Foundation for Autism)	2008	Santo Domingo	Parents	Nonprofit Organization	Active
Centro de Atención Integral para la Discapacidad-CAID (Comprehensive Disability Care Center)	2013	Santo Domingo Oeste, Santiago, San Juan de la Maguana	Office of the First Lady	Governmental	Active
Fundación Autismo en Marcha-AUMA (Autism on the Move Foundation)	2013	Santiago	^1^ INA	Nonprofit Organization	Active
Luz y Esperanza por el Autismo (Light and Hope Foundation for Autism)	2010	Santiago	Parents	Nonprofit Organization	Active
Centro Tomatis^®^ RD-Valencia	2006	Santo Domingo	^1^ INA	Private Institution	Active
Mujeres de Negro (women in Black)	INA	Santo Domingo	Parents	Nonprofit Organization	Active

^1^ INA: Information not available.

## Data Availability

Not applicable.

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
