# Peer review of "Autism Spectrum Disorder in the Dominican Republic: A Mini Review of the Current Situation"

_children, 2023, doi:10.3390/children10010121_

Round 1

Reviewer 1 Report

The review described mainly the diagnostic and treatment network in the Dominican Republic. This review may fee contribution to US and European and thus this paper is presented in the journal of Dominican Republic. 

No contribution in the field.

Author Response

We thank the reviewers for their helpful comments and suggestions. We have attempted to address each of the issues in our responses below.

“The review described mainly the diagnostic and treatment network in the Dominican Republic.”

Response: This is true. As the title indicates, the topic of this narrative review is the status of services, including for diagnosis and treatment, for families with ASD living in the DR. 

“This review may fee contribution to US and European and thus this paper is presented in the journal of Dominican Republic.”

Response: We do not understand this comment.

“No contribution in the field.”

Response: We respectfully disagree with this assessment.

“Extensive editing of English language and style required”

Response: In the absence of any context for this comment and given that the other reviewers did not share this view, we must assume that the English language aspects of this manuscript are acceptable.

Reviewer 2 Report

- Materials and Methods:

Authors also searched governmental and non-governmental institutional website of Dominican Republic, however they only included articles in English published in peer-reviewd journals: could they miss information from public government?

- Discussion: nowadays it seems there is consensus on the environment involvement in ASD, could it the same for DR?

- What about ABA as "official" intervention in DR? 

Author Response

“Authors also searched governmental and non-governmental institutional website of Dominican Republic, however they only included articles in English published in peer-reviewed journals: could they miss information from public government?”

Response: The information from the Dominican Republic website is exclusively in Spanish and then was translated into English by the lead author. It is unlikely that information was lost or misinterpreted during the translation process since Ms. Fernandez-Figuereo (originally from the DR) is English proficient. The information for this official website was used to build “Table 1”.
The original names of the institution were kept in Spanish and an English translation was added to facilitate public understanding. The methodology for building the table has been added as a part of “Material and Methods”. As for the rest of the content of the review, the articles included were in English and published in peer-reviewed journals. 

“Discussion: nowadays it seems there is consensus on the environment involvement in ASD, could it the same for DR?” 

Response: There’s no reason to conclude that DR would be any different from anywhere else, but there is not formal documentation that can support these events yet. Most of the activity around ASD in the DR up to this moment is related to parents advocating for better conditions for ASD individuals. At this point in time, this issue needs more visibility so that ASD individuals in low-middle income countries can have a better quality of life along with opportunities to participate in scientific research of these populations. 

“What about ABA as "official" intervention in DR?”

Response: ABA is used by some institutions in the Dominican Republic; however, we could not find any documentation in articles published in English. 

Reviewer 3 Report

Very interesting and valuable read. The research is definitely needed and much work needs to be carried out in the field. 

My only suggestion would be to add a few more country examples of more low income countries where ASD is more acknowledged and treatment is more available. The example of Jamaica was very interesting.  

Author Response

“My only suggestion would be to add a few more country examples of more low-income countries where ASD is more acknowledged, and treatment is more available. The example of Jamaica was very interesting.”

Response: Three more examples were added to the review. A study in Brazil, where the government provides guidelines to families and caregivers for the care of ASD individuals, a study in Tunisia, that shows evidence that parents of ASD individuals experience high level of stress in part for late diagnostic, and one from China that presents evidence that low educated mothers can be detrimental to the early detection of ASD.

Round 2

Reviewer 2 Report

I'm satisfied of authors' responses to my questions.